# Gender-Specific Significance of Peer Abuse during Childhood and Adolescence on Physical and Mental Health in Adulthood—Results from a Cross-Sectional Study in a Sample of Hospital Patients

**DOI:** 10.3390/ijerph192315986

**Published:** 2022-11-30

**Authors:** Astrid Lampe, Tobias Nolte, Marc Schmid, Hanna Kampling, Johannes Kruse, Vincent Grote, Michael J. Fischer, David Riedl

**Affiliations:** 1Ludwig Boltzmann Institute for Rehabilitation Research, 1100 Vienna, Austria; 2VAMED Rehabilitation Center, 6780 Schruns, Austria; 3Wellcome Department of Imaging Neuroscience, University College London, London WC1N 3AR, UK; 4Anna Freud National Centre for Children and Families, London N1 9JH, UK; 5Research Department for Child and Adolescent Psychiatry, University Psychiatric Hospitals Basel, University of Basel, 4002 Basel, Switzerland; 6Department of Psychosomatic Medicine and Psychotherapy, Justus Liebig University Giessen, 35390 Giessen, Germany; 7Department for Psychosomatic Medicine and Psychotherapy, Medical Center of the Philipps University Marburg, 35037 Marburg, Germany; 8VAMED Rehabilitation Center Kitzbuehel, 6370 Kitzbuehel, Austria; 9University Hospital of Psychiatry II, Department of Psychiatry, Psychotherapy, Psychosomatics and Medical Psychology, Medical University of Innsbruck, 6020 Innsbruck, Austria

**Keywords:** peer abuse, adverse childhood experiences (ACEs), health impairment

## Abstract

Peer abuse (PA) is a widespread and gender-sensitive form of Adverse Childhood Experiences (ACEs). However, research on its influence on physical and mental health in adulthood remains scarce. The aim of this study was to investigate gender-specific associations between PA and physical and mental health in adulthood in a sample of general hospital patients. A cross-sectional study at the University Hospital of Innsbruck was conducted. Data on ACEs, physical and mental health were collected using self-report questionnaires. We compared patients with no ACEs, PA only, ACEs without PA, and ACEs with PA using gender-specific binary logistic regressions to investigate the association of PA with physical and mental health. A total of 2,392 patients were included in the analyses. Women reported more emotional PA (13.1% vs. 9.4%; *p* = 0.006), while men reported more physical PA (8.3% vs. 5.2%; *p* = 0.003). PA was associated with a higher likelihood for depression (OR = 2.6), somatization (OR = 2.1), as well as worse physical health (OR = 2.1) in women but not in men. This study is the first to present data on the gender-specific detrimental effect of PA on physical and mental health in adulthood. Especially for women, PA poses a significant health risk. Thus, we should be aware of these effects and offer adequate support for affected individuals.

## 1. Introduction

A constantly growing body of evidence has documented the detrimental effect of Adverse Childhood Experiences (ACEs) on physical and mental health in adulthood [1,2,3,4,5,6,7]. Typically, ACEs include various types of physical and/or emotional ill-treatment, sexual abuse, neglect, and commercial or other exploitation experienced by children under 18 years of age [8]. More recent research findings indicate that previously understudied ACEs such as peer abuse may also contribute to the health-harming effects of ACEs [9].

Peer abuse (also referred to as ‘bullying’) is defined as the use of physical or verbal power or aggression to cause distress or to exert control over another individual [10]. This may include acts of *emotional* (e.g., direct verbal attacks; systematically spreading degrading rumors behind someone’s back; posting derogatory messages online) or *physical* (e.g., being pushed, grabbed, shoved, slapped, pinched, punched, or kicked) acts of violence by same-age children or adolescents. A cross-sectional study including 202,056 children from 40 countries reported an overall prevalence of peer abuse of 12.6%, with girls being more at risk of experiencing peer abuse than boys [10]. A recent large-scale PISA study in Germany reported a prevalence of 16% for peer abuse [11].

While peer abuse has often been downplayed as belonging to normal developmental challenges, neurobiological research finding indicate that exposure to peer abuse can negatively affect the developing stress response system, which may lead to an increased risk for poorer health and educational outcomes [12,13]. Several studies have linked peer abuse to altered cortisol level [14,15], neuroendocrine stress reaction [16,17], increased inflammation [18,19,20] and telomere erosion [21]. Iffland et al. [22] found that subjects with a history of relational peer abuse showed a more intense emotional change that was accompanied by a blunted skin conductance response after social exclusion.

In accordance with these results, peer abuse has been shown to be associated with worse academic performance and school adjustment [11,12], elevated somatic complaints [23], increased risk for depression [24,25,26,27,28], suicidal ideation [27,29], personality disorders [26], and psychosis [30] during childhood and adolescence. Furthermore, a heightened risk for psychosomatic symptoms (e.g., headaches) has been reported for affected children and young adolescents up to seven years later [31].

Previous research has documented an association of increasing risks for physical and mental health issues and increasing numbers of ACEs [1,2,3], thus indicating a dose-dependent effect of ACEs. However, Lereya, Copeland, Costello and Wolke [24] found peer abuse to be an independent risk factor for mental health problems or disorders in adolescents and young adults. There is a growing body of evidence indicating that, due to specific sensitive periods in child and adolescent brain development [32,33,34], the developmental period (“timing”) of abuse might be a significant predictive factor for later impairment. According to Khan, McCormack, Bolger, McGreenery, Vitaliano, Polcari and Teicher [27], among several confounders, peer abuse at the age of fourteen was identified as the strongest predictor of depression in young female adults.

Overall, research findings show that peer abuse is a highly prevalent problem during childhood and early adolescence and that it is associated with a broad number of psychosocial disturbances. There is also some emerging research regarding potential detrimental effect of peer abuse on physical health. However, these studies have only been conducted in young adults, and no study has yet investigated the long-lasting physical health effects of peer abuse.

The aim of the present study was to examine the specific association between peer abuse and physical and mental ill-health in adulthood. Specifically, we hypothesized that peer abuse is (I) a highly prevalent phenomenon, (II) more prevalent in patients with other ACEs, and (III) associated with decreased physical and mental health. Based on previous research findings, we also hypothesized that (IV) women are more strongly affected by peer abuse than men.

## 2. Materials and Methods

### 2.1. Sample and Procedure

Data from individuals who were inpatients and outpatients in various departments at the University Hospital of Innsbruck between October 2015 and March 2017 were included in this cross-sectional observational study. Departments used for recruiting had to meet two criteria: The heads of department agreed to join the study, and the on-site waiting periods in the respective departments were sufficiently long that patients could have enough time to complete the questionnaire. For a period of three months (per department), trained undergraduate psychology and medical students approached patients in waiting areas four mornings per week at the hospital. Patients were informed about the study, and those who gave oral and written consent completed the written questionnaires on their own; the completed questionnaires were then collected by the students. All patients received a handout with information about the study and were offered psychological support should they experience distress as a result of participation. Moreover, staff at the relevant departments were informed about the study before it began and were instructed to pay attention to potentially distressed patients. The authors assert that all procedures contributing to this work comply with the ethical standards of the relevant national and institutional committees on human experimentation and with the Declaration of Helsinki of 1975, as revised in 2008. The study design was approved by the research Ethics Committee of the Medical University of Innsbruck (AN2015-0175 351/4.18).

### 2.2. Measures

#### 2.2.1. Maltreatment and Abuse Chronology of Exposure Scale (MACE)

Adverse childhood experiences (ACEs) were assessed with the German version of the Maltreatment and Abuse Chronology of Exposure Scale (MACE) [35,36]. This scale consists of 75 items that retrospectively assess the severity of exposure to ten types of maltreatment during each year of childhood from birth up to age 18. The items were grouped into seven ACE clusters: emotional abuse (verbal or non-verbal), physical abuse, neglect (emotional or physical), witnessing violence (between parents or towards siblings), physical peer abuse, emotional peer abuse, and sexual abuse. Each cluster is introduced with a short introduction (e.g., for peer abuse: ‘*Sometimes children your own age or older do hurtful things like bully or harass you. If this happened during your childhood (first 18 years of your life) please provide your best estimates of your age at the time(s) of occurrence*’). Emotional peer abuse includes items such as ‘*Swore at you, called you names, said insulting things like you are “fat”, “ugly”, “stupid”,* etc. *more than a few times a year*’ (item 39) or ‘*Said things behind your back, posted derogatory messages about you, or spread rumors about you*’ (item 41). Physical peer abuse included items such as ‘*Forced or threatened you to do things that you did not want to do*’ (item 45) or ‘*Hit you so hard or intentionally harmed you in some way that you received or should have received medical attention*’ (item 48). The MACE provides an overall severity score and multiplicity score (number of types of maltreatment experienced) as well as subscales for each type of abuse with specific cut-off scores and has good test–retest reliability and validity [35,36].

#### 2.2.2. Brief Symptom Inventory (BSI-18)

Psychological distress was assessed with the *Brief Symptom Inventory* (BSI-18) [37], which consists of 18 items rated on a four-point Likert scale (from “not at all” to “very often”). Based on the items a mental health total score as well as three subscales (depression, anxiety and somatization) can be calculated, with higher scores indicating more distress. Sex-specific cut-off scores for the German BSI-18 global score are established [38], and good reliability and validity for the subscales and total score have been reported [37,39]. In our sample, a good internal consistency (α = 0.89) for the total score as well as for the subscales depression (α = 0.85) and anxiety (α = 0.80) was found, while the internal consistency for the somatization subscale was questionable (α = 0.68).

#### 2.2.3. Health Checklist (German Pain Questionnaire)

We used a comprehensive self-report list of diseases derived from the *German Pain Questionnaire* [40] to retrospectively assess the lifetime prevalence of disease as rated by patients. The health checklist covered eleven major physical disease clusters (i.e., cancer, cardiovascular diseases, neurological disorders, gastrointestinal diseases, metabolic diseases, musculoskeletal disorders, urogenital diseases, respiratory diseases, skin diseases, chronic pain, gynecological diseases) with one item per disease. For each category, examples of diseases and conditions were given. The overall health status was estimated by adding up the number of self-reported physical diseases.

### 2.3. Statistical Analyses

Sample characteristics and prevalence of ACEs in the sample are presented in the form of descriptive statistics. Analyses were limited to the data of individuals who provided complete data for the MACE, sex and age (see Table 1). Group differences were analyzed with independent sample *t*-tests and χ^2^-tests. Effect sizes were estimated using Cohen’s *d*, with *d* = 0.3, *d* = 0.5 and *d* = 0.8 indicating small, medium and large effect sizes, respectively [41].

The age at which abuse took place was clustered into four groups in accordance with the Austrian school system: 1–6 (pre-school age), 6–10 (primary school age), 10–14 (lower secondary school age), and 14–18 years (higher secondary school age). Since the MACE scale allows for the possibility to indicate multiple time-points of abuse, patients could potentially be placed in all four age groups. To highlight the quantitative additional effect of peer abuse as a form of ACEs, trajectories of ACEs that do and do not include peer abuse are shown for all individuals who reported any form of ACEs.

A stepwise binary logistic regression analysis was conducted in order to investigate which factors were associated with experiences of peer abuse (dichotomous variable: yes [above cut-off) vs. no [below cut-off]). In the first step, we separately tested the influence of sociodemographic variables (age, sex, living environment, highest level of education, disability) and other ACEs (emotional abuse, physical abuse, neglect, witnessing violence, and sexual abuse) on peer abuse. Significant predictors in each block were selected using the backward elimination method (likelihood ratio test) and were entered into the final model (forced entry). Odds ratios (OR) are presented with 95% confidence intervals (CIs).

To investigate the independent influence of peer abuse on physical and mental health outcomes, patients were divided into four groups: (a) no abuse, (b) peer abuse only, (c) any other form of abuse, and (d) peer abuse + at least one other form of abuse. The influence of these groups on health outcomes was estimated using logistic binary regression analyses. Patients with no reported abuse (group a) were used as reference category, and all analyses were controlled for the patients’ ages, living situation (dummy variable: living alone vs. living with someone), level of education (dummy: low vs. high level of education), and relationship status (dummy: living in a relationship vs. not living in a relationship). The number of physical disorders was dichotomized into ‘*lower number or comorbidities’* and *‘higher number or comorbidities’* (i.e., the median number of reported diseases was used as a cut-off). The analyses were calculated for men and women separately. Odds ratios are presented with 95% CIs. *p*-values < 0.05 (two-sided) were considered statistically significant. Statistical analyses were performed with IBM SPSS (v22.0, IBM Corp., Armonk, NY, USA) and SPSS AMOS (v24.0, IBM Corp., Armonk, NY, USA).

## 3. Results

Of all the patients approached (*n* = 3220), 84.1% (*n* = 2708) were willing to participate in the study. Individuals with missing data relating to ACEs, age, or sex (*n* = 401) were excluded from the analyses, resulting in a final sample of *n* = 2307 patients. Patients who reported peer abuse were younger, more often single, living alone, currently living in an urban environment, and had a higher level of education. Most frequently reported physical diseases were chronic pain (*n* = 712; 30.9%), followed by respiratory (*n* = 371; 16.1%), musculoskeletal (*n* = 362; 15.7%), neurological (*n* = 354; 15.3%), metabolic (*n* = 324; 14.0%), and gastrointestinal diseases (*n* = 306; 13.3%), cancer (*n* = 308; 13.4%), cardiological (*n* = 298; 12.9%), skin (*n* = 266; 11.5%), urogenital (*n* = 236; 10.2%), and gynecological diseases (*n* = 205; 8.9%). For details on sociodemographic data, see Table 1.

### 3.1. Prevalence of Peer Abuse and Associated Factors

Overall, 14.2% (*n* = 339) of the patients reported experiences with general peer abuse, with higher rates for emotional peer abuse (11.0%) than physical peer abuse (6.4%). Of the patients who had experienced peer abuse, 64.6% reported emotional peer abuse, 23.1% physical peer abuse and 12.2% both forms. While no sex difference was found with regard to general peer abuse (14.2% vs. 15.1%; χ*^2^* = 0.39; *p* = 0.53), we found significantly higher rates of emotional peer abuse among women (13.1% vs. 9.4%; χ*^2^* = 7.52; *p* = 0.006) as well as significantly higher rates of physical peer abuse amongst men (8.3% vs. 5.2%; χ^2^ = 9.04; *p* = 0.003). In addition, patients who had experienced peer abuse reported experiencing a significantly higher number of other forms of abuse with a medium effect size as compared to patients who had not experienced peer abuse (2.6 vs. 1.8; *t* = 6.01; *p* < 0.001; *d* = 0.58).

In the logistic regression analysis, living in an urban region (OR = 1.53; 95% CI: 1.18–1.98; *p* = 0.001), having a lower age (OR = 1.04; 95% CI: 1.03–1.05; *p* < 0.001), experiencing emotional abuse (OR = 2.75; 95% CI: 2.05–3.70; *p* < 0.001), witnessing intrafamilial violence (OR = 2.66; 95% CI: 1.87–3.80; *p* < 0.001), and sexual abuse (OR = 3.08; 95% CI: 1.96–4.85; *p* < 0.001) were associated with a higher risk for falling victim to peer abuse.

Most cases of peer abuse (95.8%) were experienced after the age of six. In our sample, there was a clear peak in the early teenage years: While 29.2% of all patients who had experienced peer abuse reported this to have taken place between the age of 6–10 years, 63.7% reported experiencing peer abuse between 10–14 years of age, while the numbers declined to 33.6% between 15–18 years of age. About half of the patients (53.5%) reported experiencing peer abuse during one of those age periods, while 37.1% reported it during two and 9.5% during all three age periods. Figure 1 shows how the number of ACEs per age year increases if peer abuse is included.

### 3.2. The Independent Effect of Peer Abuse on Physical and Mental Health Problems

For male patients, having experienced peer abuse only (in the absence of other ACEs) was not associated with an increased risk for any of the assessed health outcomes. Still, the cumulative effect of peer abuse and other forms of ACEs was not only associated with a significantly increased likelihood of depression (OR = 6.1), anxiety (OR = 5.6) and somatization (OR = 2.2), but also with a 2.6-time higher likelihood of having more comorbidities in adult life. For details, see Table 2.

While higher age was associated with more physical diseases (OR = 1.03, 95% CI: 1.02–1.04; *p* < 0.001) and lower levels of anxiety (OR = 1.01, 95% CI: 1.01–1.02; *p* < 0.001), lower level of education was associate with higher levels of depression (OR = 1.80, 95% CI: 1.19–2.72; *p* = 0.005) and somatization (OR = 1.41, 95% CI: 1.04–3.68; *p* = 0.029), and living alone with higher levels of depression (OR = 2.07, 95% CI: 1.12–3.56; *p* = 0.009).

For women, on the other hand, experiences of peer abuse only were associated with a significantly increased likelihood for depression (OR = 2.6) and somatization (OR = 2.0) and more comorbidities (OR = 2.1). The cumulative effect of peer abuse and other forms of ACEs was associated with the highest likelihood for more comorbidities (OR = 2.5), depression (OR = 5.3), anxiety (OR = 5.5) and somatization (OR = 2.3). While higher age was associated with more comorbidities (OR = 1.02, 95% CI: 1.02–1.03; *p* < 0.001) and lower levels of anxiety (OR = 1.02, 95% CI: 1.01–1.03; *p* < 0.001), not being in a relationship was significantly associated with higher levels of depression (OR = 2.09, 95% CI: 1.41–3.10; *p* < 0.001) and somatization (OR = 1.58, 95% CI: 1.14–2.17; *p* = 0.006). For details, see Table 3.

## 4. Discussion

A large and constantly growing body of literature describes the negative effect of childhood abuse on physical and mental health in adulthood [3,42]. However, the evidence base for the potentially detrimental association of peer abuse and the individuals’ health is still scarce. Thus, the aim of the present study was to present prevalence data for experiences of peer abuse collected from a broad sample of mixed general hospital patients and to investigate the independent and cumulative effect of peer abuse on the patients’ health.

In our sample, a considerable number of patients reported past experiences of peer abuse. Emotional peer abuse was reported more frequently than physical peer abuse. While for the overall peer-abuse score no sex-differences were observed, women reported significantly more emotional peer abuse than men, and men reported significantly more physical peer abuse than women. Furthermore, while peer abuse was found to have a significant independent detrimental effect on physical and mental health in adulthood, this was considerably more pronounced in women than in men. To investigate the specific influence of peer abuse on health in adulthood, logistic regression analyses were performed to compare patients (a) without ACEs, (b) with peer abuse only, (c) with other forms of ACEs, and (d) with peer abuse and other forms of abuse with regard to their physical and mental health in adulthood. These analyses were conducted for men and women separately. While ACEs were generally associated with worse physical and mental health outcomes in both samples, no significant influence of peer abuse only was found for the male sample. For the female sample, on the other hand, peer abuse was associated with a two-fold increased risk for worse physical health and somatization, a 2.6-time higher risk of depression.

Overall, approximately 14% of all patients reported experiencing peer abuse. This is in line with recent studies, which report a prevalence of 13 to 16% for peer abuse [10,11]. As we have previously reported, peer-abuse was more frequently reported by younger patients. We have discussed potential reasons for generation differences in ACE reports in more detail elsewhere [4]. In our study, a further differentiation between physical and emotional peer abuse was made. Emotional peer abuse includes direct verbal attacks but also systematically spreading degrading rumors behind someone’s back or posting derogatory messages online. Physical peer abuse, on the other hand, included experiences of direct physical attacks (e.g., being pushed, grabbed, shoved, slapped, pinched, punched, or kicked) but also threats of physical violence. While these forms of abuse have often been downplayed, recent studies show that up to one-third of teenagers who experience emotional peer abuse develop serious psychiatric disorders such as depression and even psychotic events later in life [30]. In a recent study of adolescent psychiatric inpatients in Austria, approximately 50% of the patients reported experiencing peer abuse [43]. Severe levels of emotional peer abuse can represent a substantial stress factor or even be interpreted traumatic, potentially contributing to the aggravation of pre-existing psychopathology [44,45]. Psychiatric symptoms such as depression, on the other hand, were found to mediate the relationship of emotional peer abuse and self-harming behavior [46]. Some evidence connects peer abuse to problematic coping-behaviors such as pathological internet use [47].

It has however to be critically reflected at this point, that retrospective studies on ACEs come at the cost of potential bias: for one, the magnitude of potential memory bias which may result in both underreporting and overreporting of actual experiences, cannot be fully accounted for (a comprehensive overview of influential factors was given by Baldwin et al. [48]). Secondly, the definition of ACEs is somehow socially determined and may change over time [4]. Some individuals may even lack of the understanding that their experiences were detrimental and out of the normal range of experiences. While fully acknowledging these shortcomings of retrospective assessment of ACEs, we still think it is worthwhile to try to gain a better understanding of (early) childhood experiences and later physical and mental health to facilitate both better approaches to prevention of ACE exposure as well as treatment and support for affected individuals. As of today, we have not come across any better approach to evaluate the long-time effects of ACEs on physical and mental health in later life. The MACE represents a rather sophisticated approach to capture different types of ACEs and to try to identify specifically sensitive periods in childhood. Even when accounting for potential bias, we think that more studies like ours are direly needed since it would allow meta-analyses of results and thus a potential reduction of at least some of the sources of bias.

Like the results reported by Khan, McCormack, Bolger, McGreenery, Vitaliano, Polcari and Teicher [27], our results indicate that experiences of peer abuse clearly increased by the time the patients had entered formal schooling and reached their peak in early adolescence (i.e., between 10–14 years of age). Mutlu et al. [49] observed a statistically higher rate of cortical thinning in the right temporal regions, the left temporal junction and the left orbitofrontal cortex in female adolescents than in male adolescents. The author interpreted these findings as indicating a faster maturation of the social brain areas in females, which might partly explain the higher vulnerability of girls in this developmental phase to social exclusion by peer emotional abuse. Andersen and Teicher [50] also pointed out that the hippocampus is specifically vulnerable to stress in early adolescence, which might explain the increased risk of depression and other mental disorders associated with hippocampal function in patients who have experienced peer abuse.

Researchers have previously noted that chronic exposure to stress during childhood and adolescence may lead to alterations in the nervous, endocrine and immune systems [51,52] and that this leads to alterations in the epigenome [53]. Individuals affected by ACEs also show a higher likelihood of displaying dysfunctional coping mechanisms or health-harming behaviors, such as substance abuse, sexual risk taking, or physical inactivity [3]. Some studies show that the cumulative effect of these physiological alterations and increased health-compromising behavior leads to a higher risk for impaired physical health in adulthood as well as an increased mortality in later life [1,3,54]. Previous studies also indicate that peer abuse might have a uniquely detrimental effect on physical health and mental health in adulthood [31,55,56]. However, these studies were conducted with young adults only, and it is unclear whether such detrimental effects of peer abuse continue to act throughout the entire lifespan.

More than half of the patients who experienced peer abuse in adolescence suffered from other adverse childhood experiences. Although our data do not allow us to infer causal relationships, we can assume that peer abuse and other forms of ACEs are associated. Exposure to ACEs such as physical violence, sexual abuse, or emotional neglect by the individual’s caregiver(s) may strongly compromise the development of mentalizing capacities [57]. The affected individuals are increasingly vulnerable to any critical social situation during their lives, as they often cannot easily ‘read’ the emotions, motivations, intentions, beliefs, or wishes of others or recognize their own. Because the social situations associated with peer-abuse are so highly disturbing, affected individuals can begin to habitually mistrust their environment and fail to develop epistemic trust, that is “the openness to the reception of social communication that is personally relevant and of generalizable significance” [58]. Thus, they may be prone to being isolated and excluded from the salutogenic experiences provided by social learning. This theoretical concept is empirically supported by studies such as the E-risk study [59]. In their longitudinal twin study on 2,200 children, the authors found lower mentalizing capacities in individuals who had experienced problems in peer relationships, especially after becoming a victim of bullying, acting as a bully, or switching between a bully and a victim role. Epistemic disruption has furthermore been shown to partially mediate the relationship between maltreatment and PTSD as well as complex PTSD symptomatology [6].

Thus, based on the results of our study that highlight the long-term detrimental associations of peer-abuse with physical and mental health several issues should be considered in future research. For one, we need to get better in identifying and properly reacting to children and adolescents who suffer from peer abuse. Research indicates that teachers often perceive the gravity of the abuse as less intense than affected students [60] and that only about half of the affected students had told their situation to anyone [61]. Only if peer abuse is recognized as a detrimental experience and taken seriously by society and especially teachers, we will be able to create an atmosphere in which affected individuals feel free to speak up and to search for help. It was shown that the teachers attitude towards bullying as well as and perceived peer support for victims are significant facilitators for victims to step forward [61]. It therefore seems justified to further investigate how to best implement anti-bullying programs in school and how to foster the student’s individual–but also their collective–resilience.

Secondly, more knowledge on how to best support individuals with peer-abuse experiences later in life. Peer abuse typically takes place at a time when the individual breaks away from the family of origin and begins to identify with a new social group. In this developmental phase appropriate social roles outside the nuclear family must be found and identified with, which can be seen as both an opportunity and a challenge. f this process is characterized by exclusion, rejection and devaluation, deep insecurity arises. If there is a lack of epistemic trust and mentalizing ability, which arises in an environment characterized by arbitrariness, neglect and violence, this developmental opportunity cannot be seized, epistemic mistrust, restricted social learning and further relationship difficulties are the consequences. As we have described above, it can be assumed that both the epistemic trust and mentalization ability of affected individuals may be reduced [62]. Fostering the mentalization ability and development of epistemic trust may help affected individuals to increase their social functioning [63] (including better social risk assessment) and to decrease long-lasting psychological issues [57,64]. Thus, the effectiveness of mentalization-based treatment programs for individuals who suffered from peer-abuse should be investigated in future trials. Additionally, we recommend to routinely assess experiences of social exclusion in adolescence as well as sexual or physical violence and emotional neglect as means for careful treatment planning in psychotherapy and psychosomatic treatment.

This study has several strengths: The study has been conducted in a comparably large and diverse sample of hospital patients. As we have not focused on at-risk samples, such as psychiatric or psychosomatic inpatients, the results of our study may be more clearly applicable to a general hospital population. However, this study also had some limitations: The assessment of physical and mental health was patient-rated, which is a subjective measure. However, as this approach was taken to assess subjective impairment due to disease, the findings provide a picture of the health problems that were the most important to the individual patient at the time. The main limitation of this study is that the data on ACEs were collected retrospectively. A recent meta-analysis found only limited agreement between retrospective and prospective reports of ACEs [48]. While the authors of the meta-analysis argue that retrospective assessment may be more sensitive for ACEs than prospective measures, retrospectively reported exposure is always prone to memory bias. In general, researchers assume that retrospective self-reporting leads to an underestimation of child maltreatment [65]. In addition, no information on being a bully or on switching between bully and victim roles was collected.

## 5. Conclusions

The results of our study underscore the detrimental effects of peer abuse on the physical and mental health of individuals who experienced this peer abuse in adulthood. Our results should be understood as a call to action to improve both the prevention and treatment of health issues associated with peer abuse. While peer abuse is a ubiquitous experience, our results indicate a clear pattern of gender differences: The female members of our sample were significantly more often confronted with peer abuse and also seem to experience a more pronounced, long-term impact of abuse, both physically and emotionally. Based on our findings we strongly encourage readers to recognize the role of peer abuse for future mental and physical health and thus to intervene rather than dismiss concerns of affected children, adolescents and young adults. We highlight the need for gender-sensitive prevention services, especially in schools. To ensure the long-term efficacy and effectiveness of bullying prevention programs in schools, it will be necessary to create structures with sufficient resources to implement the programs continually as part of the normal class routines [66].

## Figures and Tables

**Figure 1 ijerph-19-15986-f001:**
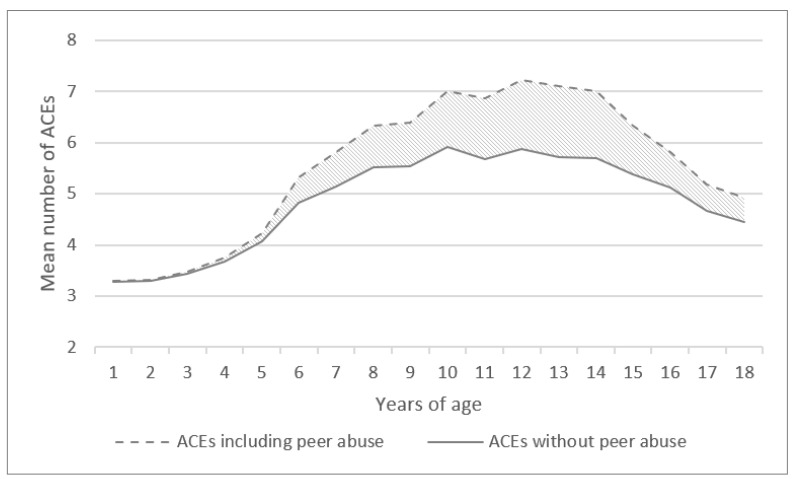
Mean number of Adverse Childhood Experiences (ACEs) per age year for patients with at least one form of ACE; hatched area marks the increase in ACEs when peer abuse is considered.

**Table 1 ijerph-19-15986-t001:** Sociodemographic characteristics (*n* = 2307).

	No Peer Abuse (*n* = 1968)		Peer Abuse (*n* = 339)			
	*n*	(%)	*n*	(%)	χ^2^	*p*-Value
**Sex**						
male	907	(46.1)	150	(44.2)	0.39	0.53
female	1061	(53.9)	189	(55.8)		
**Age groups**						
18–30 years	512	(26.0)	167	(49.3)		
30–50 years	720	(36.6)	115	(33.9)		
50–70 years	607	(30.8)	51	(15.0)		
>70 years	129	(6.6)	6	(1.8)		
**Relationship status**					41.65	<0.001
married/long-term relationship	1332	(67.7)	182	(55.2)		
single	422	(21.4)	126	(38.2)		
divorced	129	(6.6)	21	(6.4)		
widowed	28	(1.4)	1	(0.3)		
missing data	57	(2.9)	9	(2.7)		
**Level of education**					33.41	<0.001
school not finished	22	(1.1)	15	(4.4)		
compulsory school	147	(7.5)	30	(8.8)		
compulsory school and apprenticeship	722	(36.7)	89	(26.3)		
higher education	559	(28.4)	106	(31.3)		
university degree	364	(18.5)	80	(23.6)		
missing data	154	(7.8)	19	(5.6)		
**Living situation**					51.44	<0.001
living alone	354	(18.0)	92	(27.1)		
living with partner/family	1192	(60.6)	149	(44.0)		
living with family of origin	113	(6.8)	28	(8.3)		
living in shared apartment	132	(6.7)	47	(13.9)		
missing data	157	(8.0)	22	(6.5)		
**Living environment**					26.51	<0.001
urban	763	(38.8)	184	(54.3)		
rural	1085	(55.1)	141	(41.6)		
missing data	120	(6.1)	14	(4.1)		

**Table 2 ijerph-19-15986-t002:** Sex-specific adjusted logistic regression analyses for male patients (*n* = 1101), showing the influence of ACEs and peer abuse on physical and mental health outcomes.

	Overall Model Fit		No Abuse (*n* = 746)		Peer Abuse Only (*n* = 66)		Other Abuse Only (*n* = 205)		Peer Abuse & Other Abuse (*n* = 84)	
	R^2^	*p*-Value ^a^	OR	*p*-Value	OR (95% CI)	*p*-Value	OR (95% CI)	*p*-Value	OR (95% CI)	*p*-Value
physical health ^1^	9.0%	<0.001	1.0	--	1.28 (0.72–2.26)	0.40	1.10 (0.75–1.53)	0.72	2.60 (1.52–4.42)	<0.001
mental health ^2^	9.6%	<0.001	1.0	--	1.95 (0.98–3.87)	0.056	1.86 (1.20–2.90)	0.006	5.24 (3.02–9.06)	<0.001
depression	11.6%	<0.001	1.0	--	2.11 (0.98–4.54)	0.06	1.83 (1.11–3.04)	0.019	6.05 (3.39–10.82)	<0.001
anxiety	9.9%	<0.001	1.0	--	1.81 (0.97–3.36)	0.06	1.80 (1.19–2.70)	0.005	5.61 (3.30–9.54)	<0.001
somatization	3.3%	0.006	1.0	--	1.55 (0.84–2.86)	0.16	1.35 (0.91–2.00)	0.13	2.16 (1.26–3.68)	0.005

All analyses adjusted for patients’ ages, relationship status, living situation, and level of education; OR = Odds ratio; R^2^ = Nagelkerke’s R^2^ (total explained variance); ^a^ *p*-value for overall model fit; ^1^ Physical health = dichotomized variable, divided by the median number of reported symptoms (median = 2.0; 0 = ‘lower number or comorbidities’ and 1 = ‘higher number or comorbidities’); ^2^ Brief Symptom Inventory (BSI) total score.

**Table 3 ijerph-19-15986-t003:** Sex-specific adjusted logistic regression analyses for female patients (*n* = 1,291), showing the influence of ACEs and peer abuse on physical and mental health outcomes.

	Overall Model Fit		No Abuse (*n* = 782)		Peer Abuse Only (*n* = 81)		Other Abuse Only (*n* = 320)		Peer Abuse & Other Abuse (*n* = 108)	
	R^2^	*p*-Value ^a^	OR	*p*-Value	OR (95% CI)	*p*-Value	OR (95% CI)	*p*-Value	OR (95% CI)	*p*-Value
physical health ^1^	6.9%	<0.001	1.0	--	2.08 (1.25–3.48)	0.005	2.03 (1.52–2.72)	<0.001	2.49 (1.58–3.95)	<0.001
mental health ^2^	10.8%	<0.001	1.0	--	1.54 (0.84–2.80)	0.16	2.82 (2.04–3.89)	<0.001	5.16 (3.31–8.04)	<0.001
depression	10.4%	<0.001	1.0	--	2.63 (1.46–4.75)	0.001	3.54 (2.49–5.02)	<0.001	5.28 (3.29–8.50)	<0.001
anxiety	12.6%	<0.001	1.0	--	1.41 (0.79–2.52)	0.25	3.21 (2.35–4.40)	<0.001	5.46 (3.51–8.49)	<0.001
somatization	4.3%	<0.001	1.0	--	1.98 (1.19–3.30)	0.009	1.95 (1.45–2.62)	<0.001	2.32 (1.48–3.64)	<0.001

All analyses adjusted for patients’ ages, relationship status, living situation, and level of education; OR = Odds ratio; R^2^ = Nagelkerke’s R^2^ (total explained variance); ^a^ *p*-value for overall model fit; ^1^ Physical health = dichotomized variable, divided by the median number of reported symptoms (median = 2.0; 0 = ‘lower number or comorbidities’ and 1 = ‘higher number or comorbidities’); ^2^ dichotomized Brief Symptom Inventory (BSI) total score (above vs. below cut-off).

## Data Availability

The data collected in the study are available from the corresponding author upon reasonable request.

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
