# Peer review of "Gender-Specific Significance of Peer Abuse during Childhood and Adolescence on Physical and Mental Health in Adulthood—Results from a Cross-Sectional Study in a Sample of Hospital Patients"

_ijerph, 2022, doi:10.3390/ijerph192315986_

Round 1

Reviewer 1 Report

This was a fascinating study of the role of peer abuse and other ACEs on both physical and mental health. It was completed in a general hospital setting which, as noted, has advantages relative to assessments completed in psychiatric settings because of the greater confidence in generalizing the sample. This research is an important early step in setting the framework for understanding the impact of peer abuse on physical and mental health.

On the downside, this is a retrospective and cross-sectional study. Because the research is cross-sectional, we have more difficulty identifying whether the relationship is causal. Because the research is retrospective, we don’t know whether the relationship is accurate or the distortion of memory (either positive or negative) that comes with time (Baldwin et al., 2019). Some people likely gloss over or normalize abuse with time, while others may now label things as abusive that weren’t initially seen as abusive or even problematic. We might predict, though, that their current label is what would be related to outcomes. On the other hand, some people may be significantly bothered by events without ever validating their experience with a name.

As this article was written, it seems to underestimate the impact of these limitations (being hidden at the end of a paragraph at the end of the Discussion). Further emphasizing these limitations would be helpful and important.

Finally, in the Conclusions, after the first sentence, the authors might further emphasize the importance of their findings. For example, many people (in the US???) dismiss the role of peer abuse on physical and mental health. Encourage readers to recognize the role of peer abuse in future functioning and intervene rather than dismiss student concerns.

 Some editorial comments:

Line 123: It should be “you’re fat” (not your fat)

Line 124: Ditto above.

Lines 175-176: “The influence of these groups on health outcomes were…” As “were” refers to “influence,” this should be “The influence…was..”

Line 365: “Our results…” This should be plural.

Author Response

Thank you for reviewing our study and thus helping us to improve the quality of our manuscript. We have provided a detailed point-to-point response (please see attachment).

Reviewer 2 Report

The article addresses the issue of how experiencing peer abuse is related to selected indicators of physical and mental health in adults. Appreciating the potential practical value of the empirical verification of this issue, I see many shortcomings in the implementation of the research idea, including insufficient justification and indication of implications that would show the usefulness of the results. Detailed remarks:

Title

Please change the title - you cannot talk about influence, it was not checked. The term significance is more accurate.

Introduction

A relatively broad literature review. Specific and legible data allow the reader to learn about the current state of knowledge in the discussed area. Key concepts explained.

Materials and methods

1.      The research was carried out with patients of a university hospital (its various departments). It was a selected population of people who seek medical help due to various ailments, disorders, or diseases. This is the basic weakness of this research. I understand that this is a more accessible population, however, the results obtained here cannot be generalized to the general population, especially when it comes to inferring the prevalence of peer abuse or, more broadly ACEs. I suggest specifying in the title of the article that the research was conducted with a population of hospital patients.

2.      The patients gave their informed consent to participate in the research, however, as the authors write, the research was potentially distressing for the patients. I am wondering in this case, if the interest of researchers (understood as the desire to obtain material) was not more important than the well-being of the patients. This is an important ethical issue. In this context, it is particularly important to justify the purpose of the research and to show its usefulness, which unfortunately is missing from the manuscript.

3.      What surprises me is how authors dealt with the MACE tool: they treated the results of individual subscales separately (peer abuse and other forms of abuse). Whereas, certain properties confirming the reliability or validity of the scale refer to it as a whole.

4.      Respondents experienced abuse at a different age - how was this problem solved in the age-adjusted calculations (for this reason we have, for example, a sum of 120%, p. 6)?

5.      What were the comorbidity rates?

Results

The result, “Patients who reported peer abuse were younger (...)” may be caused by the fact that they remember their childhood experiences better, which is obvious considering the broad age range of the study group.

Discussion

1. Please add that there were no gender differences in the total score of peer abuse.

2. Which result indicates, “... and a two-time higher risk for living in a violent relationship” (p. 8)?

3. Considering the fact that the respondents were recruited from the patient population, the authors cannot compare the indicators obtained here with the data obtained from the general population.

4. Referring to other researchers, the authors write, “Severe levels of emotional peer abuse can represent a substantial stress factor or even be interpreted traumatic, potentially contributing to the aggravation of psychopathology” (p. 8). So it may be that peer violence is a factor exacerbating the already existing symptoms of disorders and diseases. Is that correct? This is an important statement for the design of the authors’ research and the interpretation of the results obtained in the patient population.

5. The statement, “our results indicate that experiences of peer abuse clearly increased by the time the patients had entered formal schooling and reached their peak in early adolescence” (p. 8) would be justified in longitudinal studies and with younger respondents: below six years of age.

6. What does the term “typical” hospital patients mean?

7. The authors, writing about the limitations of their own research, point to the problem with the retrospective approach to abuse. In this context, it is worth considering why many studies, as the authors write, focus on adolescents or young adults. The justification for selecting younger age groups is also that it is then possible to take earlier support and help activities that will reduce the risk of negative consequences of abuse.

8. I believe that the statement, “To our knowledge, this is the first study to examine the impact of peer abuse not only on mental but also on physical health in adults” is not justified, given that the authors themselves have already published studies on this topic (item 4), taking into account various forms of violence, including peer abuse. There are also other works dealing with this issue, which the authors themselves refer to: 19, 24, 54, 55.

The statement about the lack of research among adults considering the indicators of physical and mental health is not a good justification for the authors’ own research here, nor a confirmation of their cognitive and practical significance.

Conclusions

The authors give only perfunctory implications for practice, which is a significant weakness of this work. They do not indicate research implications.

Author Response

(The authors gave the same response as above.)

Round 2

Reviewer 2 Report

The authors clarified all doubts and made necessary additions to the article. Thank you for your positive response to the comments.